# Unleashing Creative Synergies: A Mixed-Method Case Study in Music Education Classrooms

**Mário Anibal Cardoso [1], Elsa Maria Gabriel Morgado [1]** and **Levi Leonido [2,3],***

1 Center in Basic Education, Polytechnic Institute of Bragança, 5300-253 Bragança, Portugal; cardoso@ipb.pt (M.A.C.); elsa.morgado@ipb.pt (E.M.G.M.)
2 School of Human and Social Sciences, University of Trás-os-Montes and Alto Douro, 5000-801 Vila Real, Portugal
3 Center for Research in Arts Sciences and Technologies, Portuguese Catholic University, 4169-005 Porto, Portugal
* Correspondence: levileon@utad.pt

**Abstract:** Algorithmic music composition has been gaining prominence and recognition as an innovative approach to music education, providing students with opportunities to explore creativity, computational thinking, and musical knowledge. This study aims to investigate the impact of integrating algorithmic music composition in the classroom, examining its influence on student engagement, musical knowledge, and creative expression, as well as to enhance computational thinking skills. A mixed-method case study was conducted in three Basic Music Education classrooms in the north of Portugal, involving 71 participants (68 students and 3 music teachers). The results reveal: (i) both successes and challenges in integrating computational thinking concepts and practices; (ii) pedagogical benefits of integrating programming platforms, where programming concepts overlapped with music learning outcomes; and (iii) positive impact on participants' programming self-confidence and recognition of programming's importance. Integrating algorithmic music composition in the classroom positively influences student engagement, musical knowledge, and creative expression. The use of algorithmic techniques provides a novel and engaging platform for students to explore music composition, fostering their creativity, critical thinking, and collaboration skills. Educators can leverage algorithmic music composition as an effective pedagogical approach to enhance music education, allowing students to develop a deeper understanding of music theory and fostering their artistic expression. Future research should contribute to the successful integration of digital technologies in the Portuguese curriculum by further exploring the long-term effects and potential applications of algorithmic music composition in different educational contexts.

**Keywords:** algorithmic composition; computational thinking; music composition





## 1. Introduction

Algorithmic music composition (AMC) offers an opportunity for educators to foster creativity and enhance learning experiences in the classroom [1–5]. By leveraging computational techniques, students can explore new dimensions of music composition, gain a deeper understanding of musical structures, and develop valuable problem-solving skills. AMC serves as a powerful tool for students to unleash their creativity. By using algorithms, students can experiment with diverse musical elements, such as melody, harmony, rhythm, and instrumentation, leading to the discovery of unique compositions. Algorithmic approaches provide a platform for students to explore unconventional musical patterns and break away from traditional compositional frameworks, fostering imaginative and innovative thinking. This freedom to explore and express musical ideas allows students to develop a personal artistic style while challenging conventional norms.

Regarding the development of musical understanding, AMC deepens students' comprehension of the fundamental principles of music. By engaging with algorithms and

computational techniques, students are prompted to analyze musical structures, dissect patterns, and make informed decisions about musical elements [1–3,6]. This process enhances their comprehension of concepts such as scales, chords, motifs, and form, enabling them to apply this knowledge in their own compositions. Algorithmic composition also promotes critical listening skills as students engage with diverse musical styles and evaluate the outcomes of their algorithmic creations. This heightened musical understanding enriches students' overall musical and digital literacy. On the other hand, AMC facilitates interdisciplinary learning by bridging the gap between music and technology. Students can explore concepts from mathematics, computer science, and data analysis as they develop algorithms to generate music. For example, they can experiment with mathematical patterns, explore algorithmic probability, or delve into the intricacies of data-driven composition. This interdisciplinary approach fosters connections between various subjects, stimulating intellectual growth and promoting a holistic understanding of different disciplines. Furthermore, algorithmic composition encourages collaboration, as students can work together to combine their expertise and create multi-dimensional compositions that integrate music, technology, and other domains.

Because of its unique features, algorithmic music composition is a powerful catalyst for developing students' critical problem-solving skills. As part of this innovative approach, students are presented with a variety of challenges that test their ability to identify and select appropriate algorithms, meticulously fine-tune various parameters, and carefully evaluate the resulting musical output. Through these challenges, students embark on a transformative journey. They learn to overcome obstacles and achieve their desired musical results through thoughtful problem-solving. Through this process, students develop critical thinking, logical reasoning, and adaptability (skills valuable across domains); also, they are encouraged to creatively overcome obstacles [7–10], learn from failures, and persist in their pursuit of musical excellence. This problem-solving mindset extends beyond music, preparing students for the challenges they may face in their academic and professional lives. AMC provides an inclusive environment that accommodates diverse learning styles and abilities. Students can personalize their compositional processes by adapting algorithms to suit their preferences and artistic goals. This flexibility enables students to explore their unique musical identities, experiment with different styles and genres, and encourages self-expression. Algorithmic composition also fosters independent exploration and experimentation, empowering students to engage in self-directed learning experiences. They can uncover new possibilities, take risks, and discover their own creative potential through the hands-on exploration of AMC.

This study aims to investigate and explore the impact of integrating algorithmic music composition in the classroom, examining its influence on student engagement, musical knowledge, interdisciplinary learning, and creative expression to enhance Computational Thinking (CT) skills. Furthermore, an interdisciplinary approach that combines music and programming may offer insights into how music teachers can support students' development of CT skills [11–14], thereby enhancing the successful integration of the Digital Technologies Curriculum (DTC). CT is recognized as a crucial skill due to several factors. First, it is believed to provide individuals with a competitive advantage in the modern society. Additionally, there is value in teaching interdisciplinary topics related to computing, such as artificial intelligence and algorithm design, because digital technologies are becoming increasingly important in our everyday lives. Lastly, CT emphasizes that programming includes more than just a basic understanding of language syntax. It also emphasizes problem-solving, algorithm design, abstraction, and self-reflection [15–17].

The research literature highlights the necessity for additional case studies on Computational Thinking (CT) carried out in school classrooms to enhance the comprehension of how instructors can proficiently teach and evaluate this metacognitive ability [18–22]. Previous studies on metacognition suggest that children typically begin developing metacognitive skills around the age of eight, with significant development occurring during early adolescence [22,23]. From a musical perspective, CT promotes the acquisition of skills that are

relevant to music composition. These skills include pattern recognition, abstraction, and algorithmic reasoning [13,14,24]. Prominent tools and frameworks utilized in algorithmic music and sound computing (EarSketch, Sonic Pi, Tune Pad, Max/MSP, Pure Data, Jython-Music, SuperCollider, ChucK, MATLAB, Csound, and Faust) offer excellent opportunities for the transfer and overlap of concepts, and foster learning outcomes in both music and programming languages [25,26]. The relationship between computational thinking and algorithmic music is symbiotic and holds great potential in music education.

## 2. Methods

Considering the research objectives that guide this study, a comprehensive research design has been developed to facilitate the iterative processes of (de)construction within the theoretical/technical dimension and the various elements (musical work) that emerge from empirical research. The design incorporates the principles of totality, recursion, and transformability. Totality involves perceiving the musical work as an open, dynamic, and interconnected system. Recursion permits forecasting the dialogic correlation amid its constituents, whereas transformability concentrates on cultivating the aforesaid correlation.

A mixed-methods approach was adopted to ensure comprehensive analysis, synthesizing, and generating data [27–31]. This approach combines both qualitative and quantitative methods, leveraging the respective strengths and weaknesses of each. The integration and triangulation of data from multiple sources increase the validity and reliability of the findings. While mixed-method methodologies originated in psychology, they have gained traction in educational research, forming their own distinct approach within the field [27–29,31]. Creswell [29] highlights terms such as multimethod, integrating, and synthesis, which have been used to describe this approach; furthermore, the approach has evolved to become a recognizable and distinct method of inquiry within educational research. To gather data and address the research questions, instruments and a measurement tool were designed: (i) pre-questionnaire (17-item survey designed to measure prior experiences in music composition and programming) (Section S1 in Supplementary Material); (ii) post-questionnaire (40-item survey designed to measure participants' attitudes about programming music with Sonic Pi and EarSketch across three key construct areas: pleasure, significance, and self-confidence—participants indicate their responses on a 7-point Likert-type scale (1 = very untrue of what I believe; 7 = very true of what I believe)) (Section S2 in Supplementary Material); (iii) semi-structured interviews of music teachers (background information, music composition and programming experience, teaching approaches, integration of the technology, student engagement, professional development, and future perspectives) (Section S3 in Supplementary Material); (iv) class observations; (v) class activities/tasks and student reflections (learner workbook); and (vi) a scoring/measurement tool in the form of a rubric for all final music compositions based on the rating scales of Webster (consider the presence of musical characteristics, such as rhythm, texture, timbre, harmony, and expression, and also the imaginative use of some of these characteristics) [1] and the CT dimensions (Computational Concepts, Practices, and Perspectives) of Brennan and Resnick and Brennan, Balch, and Chung [15,32]. The questionnaires and interviews for this study were conducted by the researcher. In order to gather data and insights on the subject of interest, the researcher personally administered the questionnaires and conducted one-on-one interviews with the participants. This hands-on approach ensured that the data collection process was consistent and allowed for direct interaction with the respondents, fostering a deeper understanding of their perspectives and experiences. The researcher took care to maintain a structured and unbiased approach throughout the survey and interview process to ensure the reliability and validity of the gathered information.

The methodological approach employed in this study centers around a case study of creative musical experiences conducted in three music education classrooms (2nd Cycle of Basic Education) of a public school in the northern region of Portugal during the academic year 2022–2023. There were 71 participants (68 students and 3 music teachers), with ages

between 10 and 12. In terms of gender, there were 32 female and 36 male students, along with 3 male music teachers. The research was conducted in a music classroom equipped with 25 desktop computers capable of running music creation programming platforms (e.g., EarSketch, Sonic Pi, and Tune Pad). By designing algorithms to generate musical pieces, a strong connection between music and programming is established [26,33]. For the investigation, we used EarSketch and Sonic Pi platforms.

The research plan comprised three distinct phases:

- Preparatory phase: Introduction and presentation of the work unit plan to the class. An explanation of each question in the pre-questionnaire was provided, ensuring that participants had the opportunity to seek clarification. No questions were raised during this time. The pre-questionnaire was then individually completed by all participants, including the participant music teachers, utilizing the provided link on the classroom computers. Additionally, after school, the music teachers were trained (by the researcher) for an hour on the practicalities and technical setup. The role of the music teachers was primarily focused on learning alongside and supporting the students. The involvement of all participants was notable, as they were present and actively engaged in the initial preparatory phase, which centered around data collection.
- Implementation phase: Took place each week from Monday until Wednesday, from 9:00 a.m. to 10:30 a.m. Throughout the entire unit of work, no major technical issues or interruptions were encountered. However, there were instances of student absences. To ensure continuity for absent students, they were able to catch up on missed lessons with assistance from the music teachers, their classmates, and the researcher. This involved the administration of questionnaires (students) and interviews (music teachers) to all participants. Every participant, without exception, actively engaged in the completion of the questionnaires and interviews. To gain a comprehensive understanding of the participants' experiences, post-lesson reflections were conducted for a duration of 30 min after each lesson. These reflections were approached from multiple perspectives: the researcher and the participant music teachers.
- Reflection phase: The reflections captured in the implementation phase range of valuable observations relevant to addressing the research questions. These observations encompassed engagement levels, the occurrence of meaningful discussions, and observed differences among students. The reflections played a crucial role in providing qualitative insights into the unfolding of the unit of work, enabling the study to effectively address the main research questions. It is noteworthy that all students present in each lesson completed the assigned work, questionnaires, tasks, and reflections, ensuring comprehensive data collection for analysis.

During the implementation phase, the researcher conducted the teaching while the music teachers supported the students during the lessons. As communicated by the music teachers, none of the students had special needs or language issues. The pre-questionnaire revealed the following information about the students/participants:

- Prior programming knowledge: All 68 participants in the study have prior programming knowledge of the software Scratch 2. This prior programming knowledge provides a unique context for understanding their engagement with music composition and computer-based music creation.
- Specialized musical instrument training: Among the participants, a subset of seven individuals is currently receiving specialized musical instrument training. This training focuses primarily on four instruments: piano, guitar, violin, and flute. These participants are in the process of developing advanced musical skills through instrument-specific training.
- Lack of prior experience in music composition: It is noteworthy that, according to the data collected, none of the participants reported prior experience in composing music. This lack of prior composition experience provides an interesting starting point for understanding their attitudes and experiences in this area.

- No prior experience with computer-based music creation: Similarly, none of the participants reported prior experience with computer-based music creation. This finding suggests that their exposure to music composition may be limited to traditional methods and that computer-based tools may represent a new avenue for musical exploration.

Since none of the students had prior experience with the novel programming music platforms EarSketch or Sonic Pi, no modifications or adaptations were made to the unit of work.

The work unit plan involved conducting 20 lessons from Monday until Wednesday, from 9:00 a.m. to 10:30 a.m., over a ten-week period in the academic year 2022/2023. Recognizing the inherent links between music and programming, the work unit focused on cultivating an algorithmic style of music composition through the design of lesson activities/tasks (Tables 1 and 2).

**Table 1.** Unit Plan: "Sonic Pi and EarSketch Music Programming Adventures".

| | **Lesson Title** |
|---|---|
| Lesson 1 | "SoundQuest Begins": Introduction to Sonic Pi and EarSketch |
| Lesson 2 | "Rhythmic Explorations": Understanding Beats and Rhythms |
| Lesson 3 | "Melodic Discoveries": Creating Melodies |
| Lesson 4 | "Harmony Unleashed": Combining Rhythm and Melody |
| Lesson 5 | "Looping Expedition": Looping and Repetition |
| Lesson 6 | "Variations in Tune": Creating Variations |
| Lesson 7 | "Effects Alchemy": Introduction to Effects |
| Lesson 8 | "Masterpiece Assembled": Final Composition |
| Lesson 9 | "Chord Chronicles": Exploring Chord Progressions |
| Lesson 10 | "Harmonic Heights": Crafting Harmonies |
| Lesson 11 | "Tempo Tales": Automating Tempo Changes |
| Lesson 12 | "Live Code Jam": The Art of Real-time Composition |
| Lesson 13 | "Dynamic Dialogues": Adding Expressive Dynamics |
| Lesson 14 | "Timbre Travels": Exploring Sonic Textures |
| Lesson 15 | "Beatbox Bonanza": Creating Percussive Beats |
| Lesson 16 | "Instrument Innovations": Customizing Instrument Sounds |
| Lesson 17 | "Mood Magic": Conveying Emotions through Music |
| Lesson 18 | "Song Storytelling": Music as a Narrative |
| Lesson 19 | "Remix Revolution": Remixing Existing Music |
| Lesson 20 | "Grand Finale Concert": Group Composition Showcase |

**Table 2.** Unit Plan: Learning Objectives.

| | **Learning Objectives** |
|---|---|
| Lesson 1 | - Install Sonic Pi and EarSketch.<br>- Explore the interface.<br>- Create a simple drumbeat.<br>- Listen to a basic melody. |
| Lesson 2 | - Learn about beat types and durations.<br>- Code a rhythmic pattern.<br>- Modify Tempo. |
| Lesson 3 | - Understand musical notes in EarSketch.<br>- Create a simple melody.<br>- Experiment with instruments. |
| Lesson 4 | - Combine rhythm and melody.<br>- Fine-tune timing.<br>- Explore basic chord progressions. |

**Table 2.** *Cont.*

| | Learning Objectives |
|---|---|
| Lesson 5 | • Learn about loops.<br>• Use loops for music creation.<br>• Experiment with loop types. |
| Lesson 6 | • Modify the melody for variations.<br>• Add dynamics and rests.<br>• Enhance musical expression. |
| Lesson 7 | • Explore Sonic Pi's effects.<br>• Apply effects to composition.<br>• Experiment with multiple effects. |
| Lesson 8 | • Combine all elements for a complete composition.<br>• Record and listen to your creation. |
| Lesson 9 | • Explore chord progressions and harmony.<br>• Integrate chords into composition. |
| Lesson 10 | • Fine-tune harmonies.<br>• Experiment with chord inversions.<br>• Achieve a harmonious blend. |
| Lesson 11 | • Automate tempo changes.<br>• Create dynamic tempo shifts.<br>• Understand tempo's mood effects. |
| Lesson 12 | • Learn live coding techniques.<br>• Perform compositions live.<br>• Collaborate with classmates. |
| Lesson 13–20 | • Continue exploring advanced topics in music programming, such as automation, live performance, and collaboration. Composition Project. |

Designed to invite young minds on a captivating journey of sound and code, this unit plan seamlessly blends the art of composition with the power of programming. As students embark on the adventure, they will enter the realm of Sonic Pi, a dynamic programming environment, and EarSketch, a platform that fuses code with music creation. From the very first lesson, "SoundQuest Begins", where students learn how to use the software, to the grand finale in "Masterpiece Assembled", where they weave everything they have learned into a complete musical composition, each lesson brings new challenges and discoveries.

As the unit progresses, students delve into advanced topics, exploring automation, live coding, and collaborative composition, allowing them to push the boundaries of their musical creativity. Throughout their "Sonic Pi and EarSketch Music Programming Adventures" students will not only gain proficiency in music programming but also develop a deep appreciation for the intersection of technology and art. Using code as their instrument, they will compose, experiment, and ultimately share their unique musical voices with the world.

The teaching process is carefully designed to foster creativity, encourage collaboration, and empower students to become confident music programmers. Here is a glimpse into this enriching adventure:

- Exploration and Discovery: In the early lessons, we introduce students to the software and its capabilities. We encourage them to explore, experiment, and discover the vast palette of sounds and rhythms at their fingertips.
- Hands-On Learning: We believe in learning by doing. Each lesson presents hands-on coding tasks that build skills incrementally. Students start with simple rhythms and melodies and move on to more complex compositions.

- Creative Challenges: Throughout the unit, we present creative challenges that encourage students to think critically and artistically. They learn not only to code music but also to express emotions and stories through their compositions.
- Collaboration: In lessons such as Live Code Jam and Collaborative Crescendo, students work with their peers and experience the joy of making music together. This builds teamwork and a sense of community.
- Exploration of Advanced Topics: As the unit progresses, students dive into more advanced topics, such as automation and live coding, enabling them to take their compositions to new heights.
- Continuous Feedback: The teaching process incorporates continuous feedback loops. We provide constructive feedback on students' compositions, encouraging them to refine their work and develop their unique musical voices.

During the tenth week of the study, the participant teachers took part in two extra two-hour sessions with the researcher. The objective of these sessions was to continue reviewing the methods and techniques required to execute the work unit plan.

The analysis process involved a separate examination of each measure before merging the results together. The purpose of this systematic merging was to compare and verify the consistency or inconsistency of the results. The analysis occurred in four stages: (i) preparation and organization of raw data for analysis, comprising of interview transcriptions, tasks, and project scores; (ii) utilization of analysis techniques tailored to each data type and research question; (iii) systematic data triangulation to distinguish similarities and differences; and (iv) rank the results by level of significance.

Throughout the process, regular checks were performed between the data and the researcher's interpretations to maintain a high level of accuracy. The analysis stages were non-linear and repetitive to ensure the systematic examination of the data. The analysis process involved a combination of quantitative and qualitative techniques to ensure a comprehensive exploration of the research objectives.

For the quantitative aspects of the research, a quasi-experimental single-case design was used [31]. This design was chosen to examine a new learning approach in a real teaching environment of a selected school rather than a controlled environment with a random sample. There were no cases of missing data among the participants who were present during the lessons. Missing tasks are handled by averaging previous and subsequent tasks [34]. Missing qualitative data cases were left blank [35]. All the data were imported and analyzed in MAXQDA [36–38]. One-way repeated measures analysis of variance (ANOVA) for task scores was used to analyze the quantitative data [27–31]. Correlations were also calculated between music composition and programming grades (a value from 1 to 5 assigned for statistical analysis) to determine if there was a relationship between the two subjects. Additionally, correlations between task scores and project grades were assessed to examine the connection between performance and music composition grades.

Qualitative data analysis followed a directed content analysis [39,40]. Predetermined coding schemes were utilized to categorize the data according to CT and attitude frameworks. Themes were identified through thematic analysis, focusing on emerging patterns across the data. Following the principles of convergent parallel mixed-methods design, the qualitative themes were compared with the quantitative data [27–30].

Respecting the ethical-deontological principles outlined in the Ethic Letter of the Portuguese Society of Educational Sciences, the study obtained consent from the school, participating music teachers, and participating students and their parents. Efforts were made to ensure that participants fully understood the purpose and scope of the study and that their consent was given freely without any pressure or coercion. To address potential ethical risks, the following principles and procedures were implemented: (i) special attention was given to protecting the well-being of vulnerable adolescent participants and preventing any abuse of power or coercion; (ii) access to clear study materials, informed consent forms, and the option to withdraw from the study at any time; (iii) encouraged to ask questions and voice their concerns (the researcher was available via email to ad-

dress any issues or concerns); and (iv) the participants' identities were anonymized and pseudonyms were used. All data were securely handled and stored.

## 3. Discussion

The data provided raises interesting points for discussion regarding the grades of music composition projects and the correlations observed between individual and group work. The mean and standard deviation results (Table 3) indicate that students' grades were relatively close, with a majority achieving a grade of 3/5 (students understood many CT concepts). This suggests that there was a level of consistency in the quality of work produced by students, both individually and in groups. Furthermore, the strong correlations observed between the group and individual projects for programming (correlation coefficient $r = 0.74$) and music (correlation coefficient $r = 0.82$) shed light on the relationship between collaborative and individual work in terms of incorporating CT concepts. High correlations indicate minimal variation in students' application of CT concepts when collaborating compared to their individual efforts.

**Table 3.** Individual and group programming work (final grades).

| Categories | M | SD |
| --- | --- | --- |
| (1) Individual work | 3.49 | 1.01 |
| (2) Group work | 3.37 | 0.91 |

M—mean; SD—standard deviations.

One possible interpretation of these findings is that the collaborative nature of the group projects did not significantly impact the integration of CT concepts. It appears that students were able to effectively transfer their understanding and application of CT principles from individual work to the group setting. This indicates that the collaboration process did not hinder or enhance the incorporation of CT concepts, at least as measured by the observed correlations. It is important to note, however, that these relationships provide only a limited understanding of the relationship between collaboration and the use of CT concepts. Other factors, such as group dynamics, individual contributions, and the specific nature of the music composition projects, may have influenced the outcomes.

Despite these results, in the reflections carried out by the participants during the class reflections (learner workbook), some of the students emphasize the positive impact of collaborative learning experiences on personal growth, teamwork skills, and the enriching nature of shared knowledge:

*Collaborative projects have shown me that learning isn't just about what I know, but what we can achieve together. It's a chance to share our ideas and make something amazing as a team.* (Sarah, Student)

*In our group projects, we bring together our different strengths and knowledge. It's not just about the work; it's about building lifelong teamwork skills that will help us in any career.* (Mark, Student)

*When we collaborate, we create a dynamic learning environment. We're not just learning from the teacher; we're learning from each other, which makes the subject matter come alive.* (Emily, Student)

*I used to think studying was a solitary activity, but collaborative learning has shown me that there's power in numbers. Together, we can achieve more than I could on my own.* (Alex, Student)

*Collaboration isn't just about academics; it's about connecting with people, sharing experiences, and realizing that we're part of a bigger learning community.* (David, Student)

*I used to be shy about sharing my thoughts, but collaborating with my classmates has boosted my confidence. It's a safe space to express myself and learn from others.* (Carlos, Student)

> *Collaboration isn't always easy, but it's where we learn some of our most valuable life lessons—communication, compromise, and the joy of achieving something as a team.* (Mia, Student)

Further discussion could explore the potential benefits and challenges of collaborative work in music composition projects. Collaborative efforts may offer opportunities for students to share ideas, pool resources, and develop teamwork skills. On the other hand, individual work allows for personal exploration and creative expression. It would be interesting to investigate whether different group compositions, roles, or dynamics influence the integration of CT concepts differently. Additionally, future research could delve into the specific CT concepts that students employed in their individual and group projects. Understanding which concepts were utilized more effectively in collaborative settings could inform instructional approaches and support the development of CT skills in music composition. In conclusion, the data suggest that there were no significant differences in the application of CT concepts between individual and group projects in the context of music composition. This finding invites further exploration and discussion on the implications of collaboration and the effective integration of CT concepts in music education.

The results of the study shed light on the implications of CT concepts in understanding the connections between music and programming language using platforms such as EarSketch or Sonic Pi. It was discovered that in order to support interdisciplinary skills and knowledge transfer in music, the concepts of sequences, loops, parallelism, and data are essential. However, there was a concerning gap in the evidence gathered regarding the CT concepts of conditions and operators, indicating a weakness in the implementation of these concepts by beginner students in both programming and music composition. To address this gap, educators are recommended to utilize the successful CT concepts identified in the study when designing the curriculum for beginner students. Emphasizing the use of descriptive variable names can enhance student understanding and clarity in their programming and music compositions. It is important to note that conditional loops are not possible in Sonic Pi, and this should be considered when planning learning outcomes. Additionally, educators should anticipate that the concepts of conditions and operators may pose challenges for beginners using EarSketch or Sonic Pi and consider incorporating the teaching of these concepts using another programming language. In addition, the study highlights the limited qualitative evidence of students' engagement with the CT practices of 'testing and debugging' and 'reusing and remixing'. Teachers should promote a step-by-step and iterative approach for composing music, discourage a trial-and-error approach for solving problems, emphasize code comments, fix syntactical bugs, promote reuse and remix, and foster skills in programmatic linking of code components. While further research is needed to refine the strategies for teaching, monitoring, and assessing CT practices at the beginner level, implementing these recommendations can support the development of CT practices and improve learning outcomes for students in music and programming education. Educators need to carefully consider the challenges and implications outlined in the study to create a comprehensive and effective learning experience for their students.

> *While coding music can be fun, there's a gap in our understanding of how to effectively 'test and debug' our compositions. I believe that a more structured approach will help us avoid frustrating trial-and-error moments.* (Olivia, Student)

> *I've noticed that 'reusing and remixing' existing code snippets can be challenging. Teachers can guide us on how to find and adapt code components, making it easier to create unique compositions.* (Noah, Student)

> *Promoting 'reuse and remix' is fantastic! It allows us to build on the work of others and encourages creativity. Teachers should teach us how to do this responsibly and ethically.* (Ava, Student)

*Learning to 'programmatically link code components' is like learning to compose a musical masterpiece. Teachers can help us understand the power of combining code in harmony.* (Ethan, Student)

Regarding the participants' attitudes (pleasure, significance, and self-confidence), the results provide:

- Valuable insights into how engaging in the creative activity of composing through EarSketch and Sonic Pi influences participants' attitudes (pleasure, significance, and self-confidence) about programming and music composition (Table 4).
- A demonstration of the significant enhancement in participants' attitudes, indicating a more positive outlook and contributing to the existing literature by providing a comprehensive understanding of the positive influence of algorithmic music composition on attitudes towards programming.
- A quantitative analysis of each subscale that aligns with previous research and offers insights into the specific challenges educators may encounter within each subscale. However, the absence of critical responses in students' reflections raises questions about the effectiveness of the instruments and methods used to encourage deeper reflection. It suggests that the current approach may not have effectively captured the full range of participants' perspectives and experiences. Future research could explore alternative methods or instruments to elicit more critical and reflective responses from students.

**Table 4.** Participants' attitudes about programming and music composition.

| Categories | *t*-Value | Cohen's d | *p*-Value |
| --- | --- | --- | --- |
| programing | | | |
| (1) pleasure | 7.66 | 1.19 | |
| (3) significance | 10.13 | 1.57 | $p < 0.001$ |
| (5) self-confidence | 7.81 | 1.46 | |
| composition | | | |
| (1) pleasure | 6.18 | 1.35 | |
| (2) significance | 6.34 | 1.43 | $p < 0.001$ |
| (3) self-confidence | 7.08 | 0.67 | |

It is noteworthy that all students achieved high grades and task scores in both music and programming, indicating a positive relationship between academic performance and attitude. While this may explain the absence of critical voices in the student's reflections, it is important to recognize that the creative activity of music composition with EarSketch and Sonic Pi can still contribute to the development of positive attitudes toward programming. This perspective is reinforced by participating teachers:

*Music composition with tools like EarSketch and Sonic Pi isn't just about creating melodies; it's about creating a passion for programming. It's incredible to witness how these activities can ignite students' interest in coding.* (Teacher A)

*Incorporating creative music composition into programming education isn't just about teaching technical skills; it's about nurturing a love for problem-solving and innovation. EarSketch and Sonic Pi are invaluable in this journey.* (Teacher B)

*Music composition activities with EarSketch and Sonic Pi can make coding more accessible, enjoyable, and meaningful, fostering a positive attitude towards this vital skill.* (Teacher C)

These results highlight the potential of integrating EarSketch/Sonic Pi and similar tools into programming courses and music classes, particularly for beginner students with an interest in music composition.

Based on the results, several recommendations are proposed for music teachers. First, educators are encouraged to incorporate EarSketch and Sonic Pi into programming courses or music composition classes, leveraging their potential to foster positive attitudes toward

programming. Offering opportunities for both individual and group projects can enhance students' enjoyment and engagement in programming activities, as they can choose the working style that aligns with their preferences.

Moreover, while emphasizing the future employment opportunities associated with programming skills, it is essential to highlight the intrinsic benefits of learning programming as an empowering activity for self-expression. Positive attitudes can be developed by nurturing students' intrinsic interest in programming through interest-based projects, such as music composition using EarSketch and Sonic Pi. Lastly, it is crucial to emphasize to students that programming is a skill that anyone can learn, regardless of their initial proficiency. Because of its accessibility and user-friendly interface, Sonic Pi is recommended as an effective tool for beginner students at the school level to develop their programming skills with ease.

By leveraging the creative potential of music composition with EarSketch and Sonic Pi, educators can foster positive attitudes about programming among their students. This approach not only enhances students' enjoyment and self-confidence but also highlights the intrinsic benefits of programming as a means of self-expression and empowers them to explore future opportunities in the musical field. While the research instruments did not exhibit any apparent issues, and the directed content analysis approach, which utilized a modified theoretical framework for examining attitudes, presented no challenges, the unexpected lack of negative responses during the analysis of qualitative data is noteworthy. Factors such as students' high grades, above-average support, and the novelty associated with creating music using EarSketch and Sonic Pi likely contributed to the overwhelmingly positive responses. Future research could explore strategies to elicit a more diverse range of responses and capture a broader spectrum of attitudes about programming in the context of algorithmic music composition in the classroom.

This study highlights the potential of algorithmic music composition by using Sonic Pi and EarSketch to positively influence attitudes about programming. The results support the integration of algorithmic music composition into programming education, providing educators with valuable insights and recommendations to enhance student engagement, enjoyment, and self-confidence in programming activities.

The positive outcomes of this study suggest that algorithmic music composition can be a powerful tool for enhancing students' attitudes toward programming. To bring these programs to scale and make them accessible to a broader student population, several steps can be considered:

- Curriculum Integration: Integrating algorithmic music composition modules into existing computer science or programming courses can be an effective strategy. This allows students to explore creative aspects of programming alongside traditional coding skills.
- Teacher Training: Educators need training to effectively facilitate algorithmic music composition activities. Workshops and professional development programs can equip teachers with the necessary skills and knowledge to implement these activities in their classrooms.
- Resource Development: The development of educational resources, including lesson plans, tutorials, and sample projects, can help standardize the implementation of algorithmic music composition programs in various educational settings.
- Collaboration with Music Departments: Collaboration between computer science and music departments can foster interdisciplinary approaches to algorithmic music composition. This can lead to the creation of specialized courses or workshops.

Facilitators, whether teachers or instructors, play a critical role in the success of algorithmic music composition programs. They need to be trained in various aspects:

- Technical Proficiency: Facilitators should have a solid understanding of the tools used for algorithmic music composition, such as Sonic Pi and EarSketch. This includes proficiency in coding and music theory.

- Pedagogical Skills: Training should also focus on pedagogical approaches that engage students in creative coding and music composition. Facilitators should learn how to create a supportive and inclusive learning environment.
- Assessment Strategies: Developing effective assessment strategies is crucial. Facilitators should understand how to evaluate students' progress and provide constructive feedback.
- Adaptability: Facilitators should be adaptable and able to tailor their teaching methods to the needs and interests of their students. Algorithmic music composition can find a place in various educational contexts:
- Computer Science Courses: Integration into computer science courses can expose students to the creative side of programming, making the subject more engaging and enjoyable.
- Music Education: In music education, algorithmic composition can help students explore innovative ways to create music and understand the technical aspects of music production.
- Interdisciplinary Programs: Algorithmic music composition can be part of interdisciplinary programs that merge arts and technology, fostering a well-rounded education.
- Extracurricular Activities: Offering algorithmic music composition as an extracurricular activity can encourage students to explore programming in a fun and creative way outside of regular coursework.

Future research is needed to explore the long-term effects and scalability of algorithmic music composition in different educational contexts. This research can focus on:

- Long-Term Impact: Assessing how algorithmic music composition influences students' long-term interest in programming and music.
- Diversity and Inclusion: Investigating the accessibility of these programs to a diverse student population and addressing potential barriers.
- Cross-Curricular Benefits: Exploring how algorithmic music composition impacts other aspects of learning, such as problem-solving skills and creativity.

The results of this study provide a compelling case for the integration of algorithmic music composition into programming education. To realize the full potential of these programs, it is essential to address the scaling, facilitator training, and implementation aspects while also continuing to advance research in this field. Algorithmic music composition can not only enhance student engagement, enjoyment, and self-confidence in programming but also offer a creative and interdisciplinary approach to education.

### 4. Conclusions

The integration of digital technology played a fundamental role in the development of creative musical experiments in the classroom [8–10,41–43]. Digital technology served as an indispensable tool throughout the musical composition process, enriching the students' creative endeavors in our study. The results demonstrate that integrating computer programming in the music composition process fosters the exploration and expansion of new languages and codes. Algorithmic music composition offers a wealth of benefits in the educational setting. By integrating algorithms and computational techniques into music education, teachers can foster creativity, develop a deeper understanding of music, promote interdisciplinary learning, promote computational thinking, and enhance problem-solving skills.

Embracing algorithmic music composition in the classroom not only arms students with valuable skills for the digital age but also nurtures their artistic expression, cultivates a lifelong love for music, and prepares them to be versatile and adaptable in an ever-changing world. By harnessing the potential of algorithmic music composition, teachers can unleash creative harmonies by inspiring the next generation of musicians.

**Supplementary Materials:** The following supporting information can be downloaded at: https://www.mdpi.com/article/10.3390/app13179842/s1, Section S1: Questionnaire; Section S2: Questionnaire; Section S3: Interview Guide.

**Author Contributions:** Conceptualization, M.A.C.; Methodology, M.A.C.; Formal analysis, E.M.G.M.; Investigation, M.A.C.; Writing—original draft, M.A.C.; Writing—review & editing, E.M.G.M. and L.L.; Visualization, E.M.G.M.; Supervision, L.L. All authors have read and agreed to the published version of the manuscript.

**Funding:** This work has been supported by FCT—Fundação para a Ciência e Tecnologia (national public agency that supports research in science, technology, and innovation) within the Project Scope: UIDB/05757/2020.

**Institutional Review Board Statement:** Not applicable.

**Informed Consent Statement:** Informed consent was obtained from all subjects involved in the study.

**Data Availability Statement:** All data generated or analyzed during this study are included in this published article.

**Conflicts of Interest:** The authors declare no conflict of interest.

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
