# Peer review of "Unleashing Creative Synergies: A Mixed-Method Case Study in Music Education Classrooms"

_applsci, doi:10.3390/app13179842_

Round 1

Reviewer 1 Report

Interesting paper / study on the use and benefits of computer programming to help teach music composition to beginning students.  Several powerful moments, and offering of strong support for the integration of computer programming and music composition.

Overall a very nice paper.  It is time that music educators engage with 21st century technologies and ideas, similarly to how their predecessors engaged with earlier technologies (such as piano, audio recording, mathematics, amplified / electronic instruments, etc.).  

Following are some points / suggestions for improvement:

The title threw me off - "Unleashing Creative Harmonies:" - I expected the paper to be about teaching music harmony (counterpoint, chords, etc.) to music composition students.  It is not.  This is confusing.  Consider "Synergies", or "Opportunities" or something along those lines - "Harmonies" is not the right word in English.

"The distinctive advantage of Sonic Pi lies in its ability to harness the computational power of the computer to execute algorithms, enabling the creation of music compositions" - This reads like it was quoted from a Sonic Pi paper / documentation.  However, this is not a distinctive advantage, as it is shared by other music programming environments for education, such as EarSketch (which you quote), and JythonMusic (which you do not).

Provide questionnaire(s) used in an Appendix.  Very useful for others to evaluate the approach, and also to replicate it.

[Page 4]: "digital technologies teacher" - First mention of this.  Perhaps introduce earlier.  Is this the same as the researcher(s)?  Unclear."

[Page 5 - middle to bottom of page]:  Perhaps reduce some of this verbiage.  A bit too detailed, at times feeling repetitive almost.  The bulleted part seems unnecessary, for the paper. 

[Page 6 - last paragraph]:  Convert to a bulleted list.  So much important info is hiding here.  Almost every sentence is very impactful, but may be missed.  Emphasize / highlight!!! 

I wish there was less discussion and more raw data - i.e., individual (anonymized) grades across assignments, a description of the assignments used, etc.  

Here you report a lot about the study and methodology, but not enough for someone to possibly analyze your data / results differently, or - at least - lay eyes on them, in case they see something new.  Also, to be able to replicate, or adapt / improve your study. 

Again, remove some of the discussion / verbal analysis of results, and introduce more info / details on the experiment itself, such as (a) tasks students were asked to complete throughout the study, and (b) more raw data - (not just higher-level summaries).

"planning learning outcomes" - Please provide learning outcomes used - VERY important.

"using another programming language" --> again, take a look at JythonMusic.  You might find it is better suited to teaching programming via musical tasks.

[Page 8]:

"Moreover, while emphasizing the future employment opportunities associated with programming skills, it is essential to highlight the intrinsic benefits of learning programming as an empowering activity for self-expression." - a most powerful moment of the paper, currently hiding in the middle of a paragraph.  Consider using new paragraphs when something new or important is being introduced.

"specifically using Sonic Pi" - Why not EarSketch?  Please provide student tasks / programming and/or music composition assignments.  What were the students required to do?  This is a very important part of your study, which is currently lacking (not being mentioned).

"enriching the students’ creative endeavors." --> "enriching the students’ creative endeavors, in our study."

"Embracing algorithmic music composition in the classroom not only arms students with valuable skills for the digital age but also nurtures their artistic expression, cultivates a lifelong love for music, and prepares them to be versatile and adaptable in an ever-changing world." - again, a VERY powerful moment of the paper.  It should probably be in its own paragraph, with the next and final sentence.

"unleash creative harmonies" - what does this mean?  Unleash creative potential, perhaps? (Also see earlier comment about paper title).

Overall a great paper, with great impactful moments.  In summary, introduce the musical / programming tasks you requested the students to perform.  This is very important - otherwise we cannot truly appreciate the breath / impact of your study.  Also, see other points made. 

I really enjoyed reading this paper, and look forward to its final version.  I will definitely use it as a reference when discussing the importance of introducing computer programming in music educational settings.  Thank you for writing it.

Paper needs some proofreading by a native speaker of English.

Generally, it is a good idea to introduce new paragraphs for new ideas.  Manytimes, I find that important new concepts, or changes in narrative or results, are hidden inside a long paragraph.  Consider breaking up some of paragraphs into smaller units.

Suggested corrections:

[Page 1 - Abstract]:

"and creative expression, to enhance computational thinking skills" --> "and creative expression, as well as to enhance computational thinking skills"

"71 participants (68 students and 3 Music Teachers)" - why capitalize "Music Teachers"?  Do not.

" (ii) highlights the pedagogical benefits of the integration of programming platforms" --> " (ii) pedagogical benefits of integrating programming platforms"

"music composition in the classroom setting positively influences" --> "music composition in the classroom positively influences"

[Page 2]:

"a transformative journey. Learn to overcome" --> "a transformative journey. They learn to overcome"

" (skills valuable across domains), encouraged" --> " (skills valuable across domains); also they encouraged"

"Another point to highlight is your preponderance to encouraging learning and exploration."  What does this mean?  Is preponderance the right word here?  Please rephrase.

"This study aims to investigate and explore impact" --> "This study aims to investigate and explore the impact"

"knowledge of syntaxes, emphasizing problem-solving, algorithms," --> "knowledge of language syntax, emphasizing problem-solving, algorithmic design,"

"CT highlights the development" --> "CT supports (or promotes) the development"

"Prominent tools and frameworks ... offers excellent opportunities for transfer and overlap of concepts, fostering" --> "Prominent tools and frameworks ... offer excellent opportunities for transfer and overlap of concepts, and foster"

[Page 3]:

"Creswell [29] highlights those terms such as multimethod, integrating, and synthesis have been used to describe this approach, but it has" --> "Creswell [29] highlights terms, such as multimethod, integrating, and synthesis, which have been used to describe this approach; furthermore, the approach has"

"2022/2023." --> "2022-2023." ("/" indicates option, whereas "-" indicates range)

"The participants consisted of 71 participants (68 students and 3 Music Teachers) from a Music Education classroom, including 32 female" --> "There were 71 participants (68 students and 3 music teachers) from a music education classroom.  In terms of gender, these were 32 female"

"Sonic Pi code" --> "Sonic Pi programming"

"Additionally, all music teachers’ participants" --> "Additionally, all music teacher participants"

"Furthermore, music teacher, along with the school, informed the staff, student participants, and their parents" - hard to understand, rephrase.

[Page 4]

"Additionally, an hour after school was spent with music teachers, where they were familiarized with the practicalities and technical setup" --> "Additionally, after school, the music teachers were trained for an hour to the practicalities and technical setup"

"Involved the administration" --> "This involved the administration"

[Page 7]:

"programming courses and music class" --> "programming courses and music classes"

"programming courses or music composition class, leveraging its potential" --> "programming courses or music composition classes, leveraging their potential"

[Page 8]

" integrating programming training/language in music composition process" --> " integrating computer programming in the music composition process"

Author Response

We would like to thank the reviewers for their important suggestions for changes, which have been complied with (attached). 

As we can only submit one document, we have opted to add the body of the text and its appendices. 

Reviewer 2 Report

I don't know if you are receiving this again. I clicked submit and my comments here disappeared. Here they are again.

I think the paper's subject is very important and the education community can benefit a lot from work like this. However, the paper lacks too much information for the community to be able to judge its value and use its results. 

Here are some examples of missing information or questions that should be clarified. They are not an exhaustive descriptions, just examples to make evident the lack of detail:

Who are the participants? What age? What is a "music education classroom"?

Who provided the training? What is their level of expertise? 

Is there a curriculum available, or will be?

What were the specific classroom activities? Provide details and examples.

Regarding assessment:

What specific survey items were used?

What grades is the paper referring to? How were the grades assigned? By whom?

How were the questionnaires or interviews? Who carried them out? What kind of things did the participants say? Provide specific examples and quotes.

The following is poorly written or confusing "(i) prior programming experience on software Scratch". Did all the participants have prior experience? Some of them?

Regarding the discussion: It would be good to see a more thorough discussion of how these results could impact education. For example, what is needed to bring these kind of programs to scale? How should facilitators be trained? In what classes could activities like this be implemented. This is not strictly necessary, but it would improve the significance of the work.

I am recommending to accept with minor revisions, because I am guessing that the authors can improve the paper without carrying out new experiments. However, I think the issue of the quality of the description is very serious and the paper should not be published like this. It is in the best interest of the authors to improve the description.

Also, the paper is repetitive and provides way too much detail about things that are not that relevant, such as the participant consent.

The English is generally good. It requires minor revisions. I think the paper should change substantially so it is pointless to mention them here.

Author Response

(The authors gave the same response as above.)
